# Stepwise Reduction of Mycophenolate Mofetil with Conversion to Everolimus for the Treatment of Active BKV in Kidney Transplant Recipients: A Single-Center Experience in Vietnam

**DOI:** 10.3390/jcm11247297

**Published:** 2022-12-08

**Authors:** Truong Quy Kien, Nguyen Xuan Kien, Le Viet Thang, Phan Ba Nghia, Diem Thi Van, Nguyen Van Duc, Do Manh Ha, Nguyen Thi Thuy Dung, Nguyen Thi Thu Ha, Vu Thi Loan, Hoang Trung Vinh, Bui Van Manh, Hoang Xuan Su, Tran Viet Tien, Lionel Rostaing, Pham Quoc Toan

**Affiliations:** 1Department of Nephrology, Military Hospital 103, Vietnam Military Medical University, Hanoi 100000, Vietnam; 2Transplant Centre, Military Hospital 103, Vietnam Military Medical University, Hanoi 100000, Vietnam; 3Department of Endocrinology, 108 Military Central Hospital, Hanoi 100000, Vietnam; 4Center of Emergency, Intensive Care Medicine and Clinical Toxicology, Hanoi 100000, Vietnam; 5Institute of Biomedicine and Pharmacy, Vietnam Military Medical University, Hanoi 100000, Vietnam; 6Department of Infectious Disease, Vietnam Military Medical University, Hanoi 100000, Vietnam; 7Nephrology, Hemodialysis, Apheresis, and Kidney Transplantation Department, Grenoble University Hospital, CEDEX 9, 38043 Grenoble, France

**Keywords:** BK polyomavirus, immunosuppression, everolimus, renal allograft rejection, kidney transplantation, BKV genotypes, Vietnam

## Abstract

Background: No specific antiviral drug can effectively treat BKV reactivation after kidney transplantation. Thus, we evaluated stepwise-reduced immunosuppression to treat BKV reactivation. Methods: 341 kidney-transplant recipients were monitored for BKV infection (BKV-viremia, BKV-viruria). Positive samples with a significant virus load were nested PCR-genotyped in the VP1 region. In 97/211 patients presenting BKV viremia ≥10^4^ copies/mL and/or BKV viruria ≥10^7^ copies/mL, or BKV-nephropathy immunosuppression (i.e., mycophenolate mofetil [MMF]) was reduced by 50%. If viral load did not decrease within 28 days, MMF dose was further reduced by 25%, although calcineurin-inhibitor (CNI) therapy remained unchanged. If BKV viral load did not decrease within another 28 days, MMF was withdrawn and replaced by everolimus combined with reduced CNIs. Results: Only 41/97 BKV (+) cases completed the 6-month follow-up. Among these, 29 (71%) were in the BKV-I group and 12 (29%) were in BKV-IV. BKV viruria and BKV viremia were significantly decreased from 9.32 to 6.09 log10 copies/mL, and from 3.59 to 2.45 log10 copies/mL (*p* < 0.001 and *p* = 0.024, respectively). 11/32 (34.4%) patients were cleared of BKV viremia; 2/32 (6.3%) patients were cleared of BKV in both serum and urine, and 9/9 (100%) only had BKV viruria but did not develop BKV viremia. eGFR remained stable. No patient with BKV-related nephropathy had graft loss. There was a significant inverse relationship between changes in eGFR and serum BKV load (r = −0.314, *p* = 0.04). Conclusions: This stepwise immunosuppressive strategy proved effective at reducing BKV viral load in kidney transplant recipients that had high BKV loads in serum and/or urine. Renal function remained stable without rejection.

## 1. Introduction

BK polyomavirus (BKV) is a non-enveloped DNA virus first discovered in the urine of a kidney transplant recipient in 1971 [1]. BKV viruria is found in approximately 60% of kidney transplant recipients, while BKV viremia is seen in up to 13% of kidney transplant recipients, and BKV-associated nephropathy (BKVN) is found in up to 10% [2,3]. In the US, 5.7–7.5% of renal allografts are lost due to BKVN [4]. According to polymorphisms in VP1 and NCCR, BKV strains are divided into four genotypes, with the frequency of genotype I at around 80% and of genotype IV at 15% [5]. The kidney allograft biopsy is the backbone for BKVN diagnosis, severity assessment, and concomitant process evaluation. However, because biopsy is invasive and sampling error can occur, a theoretical diagnosis can be performed based on the presence of significant BKV viremia. In order to recognize post-transplant BKV replication early, screening tests in urine or plasma are recommended, thus allowing intervention and avoiding progression to BKVN or allograft rejection [4,5,6]. A reduction in the intensity of immunosuppression is the overarching principle for treating BKV viremia and BKVN. There is no specific therapeutic agent available to treat this virus-associated disease, with many agents lacking conclusive efficacy in reducing viral loads. Multiple protocols have been developed to reduce immunosuppression, but trials are yet to be conducted to compare their effectiveness [4,6,7]. However, a risk of rejection may arise when reducing the dose of immunosuppressants [8,9]. EVL is thought to play a key role in BKV control; however, its use may increase the risk of proteinuria and dyslipidemia [7,10]. In a resource-limited country such as Vietnam, our study aimed to evaluate in the setting of BKV viremia or BKVN a stepwise approach for reducing immunosuppression by first reducing only mycophenolate mofetil (MMF) exposure and, if not efficient, then discontinuing MMF and replacing it with everolimus (EVL combined) in combination with CNI-dose minimization at that stage. To the best of our knowledge, this is the first study on this topic conducted in a resource-limited country.

## 2. Materials and Methods

### 2.1. Study Design and Patients

The study was conducted on 341 kidney transplant (KTx) recipients in whom BKV was regularly assessed in both the blood and the urine at the same time (Figure 1). The BK virus was detected using real-time PCR. Significantly positive BKV samples were used for gene sequencing. There were 214 patients who tested at least once for BKV reactivation; of these, at least once serum BK viremia was < 10^4^ copies/mL and BK viruria < 10^7^ copies/mL in 114 patients. These patients underwent no intervention. Conversely, there were 97 patients with indications for therapeutic intervention because they have a high load in the serum (≥10^4^ copies/mL, *n* = 45) and/or the urine (≥ 10^7^ copies/mL, *n* = 42) or BKVN (*n* = 10). They were followed up for six months after diagnosis. At baseline, all these 97 patients were on standard immunosuppression comprising CNI, MMF, and steroids. Some patients (*n* = 17) underwent kidney allograft biopsy because of creeping serum creatinine, i.e., an impaired renal function defined as having a decrease in estimated glomerular filtration rate (eGFR) of more than 20% as compared to baseline values. The patients’ doses of immunosuppressive drugs were adjusted according to a uniform protocol (see below). All pertinent data, namely demographics and laboratory parameters, as well as kidney-transplant-related information, were collected for each patient at the time of BKV diagnosis (T0), i.e., when the patients were enrolled. BKV viremia, BKV viruria, tacrolimus and cyclosporine trough levels, serum creatinine levels, and proteinuria were recorded at least once a month during the 6 months following BKV diagnosis. Estimated GFR was calculated using the CKD-EPI formula. The data were analyzed at three periods where T0 was defined as BKV diagnosis, T1 at the 3rd month, and T2 at the 6th month after BKV diagnosis.

Regarding maintenance immunosuppression, all patients were on mycophenolate mofetil 1 g twice a day, low steroids doses, and tacrolimus. Tacrolimus trough levels were between 8 and 10 ng/mL for the first 6 months post-transplant, 7–10 ng/mL between month 6 to 12 and 5–8 ng/mL beyond the first year post-transplant.

This prospective study was conducted with the permission of the Ethics Committee of our hospital (protocol code 2885 and date of approval: 7 July 2020). All the patients gave their informed consent to participate in the study.

### 2.2. Quantification of BKV Viral Load in Urine and Plasma Samples

We used an in-house quantitative real-time PCR assay, which was developed in our laboratory with a slight modification from the previously published protocol on the Rotor-Gene Q5 plex MDx platform (Qiagen, Germany) for quantification of BKV load in urine and plasma [11,12]. The BK virus load was expressed in BKV genome copies per milliliter of urine or plasma. The lower limit of BK viral load detection at our center is 250 copies per milliliter.

Regarding BKV viremia and BKV viruria, we defined a clinical response when the decrease between 2 assessments was greater than 1 log copies/mL. Conversely, viremia/viruria were characterized as unchanged when the decrease was less than 1 log copies/mL. Finally, an increase in viremia/viruria was defined by an increase of more than 1 log copies/mL.

### 2.3. Genotyping of BKV

Primers were designed for the VP1 region of BKV, as described previously [8,9]. The external primer pair, BKS + BKAS, and the internal primer pair, BKF + BKR1, amplified a 580-bp and a 327-bp DNA fragment, respectively (Table 1). The PCR reaction was optimized in a total volume of 20 μL using a PCR kit (GoTaq Mastermix, Promega, Madison, WI, USA) containing 5 μL of DNA template, 1 μL primer, 4 μL of molecular-grade water, and 10 μL of 2X Buffer. Cycling conditions were 95 °C for 5 min, followed by 35 cycles of 94 °C for 30 s, 58 °C for 30 s, and 72 °C for 30 s, and a final extension at 72 °C for 7 min. For the nested PCR, 3 µL of the products from the PCR, using the external primer pair, was added as the DNA template for the second round of PCR using the internal primer pair. PCR amplifications were performed using the parameters described above. All reactions were implemented on an Eppendorf™ Mastercycler™ pro PCR System. Amplification products were separated by electrophoresis on 1.2% agarose and visualized under UV light after staining with ethidium bromide (10 mg/mL) and sequenced using a 3130xl sequencer.

### 2.4. Kidney Allograft Biopsy

Kidney allograft biopsies were performed in patients having allograft dysfunction, i.e., a decline in eGFR of more than 20% as compared to baseline values. BK virus nephropathy was characterized by tubular atrophy and fibrosis with a variable inflammatory lymphocytic infiltrate. BK virus-associated nephropathy was confirmed using immunohistochemical nuclear staining with anti-SV40 antibody. Concomitant acute rejection was evaluated using the Banff 2017 scoring system.

### 2.5. Immunosuppressive Protocols

All patients in the intervention group were treated with a stepwise decrease in immunosuppression. First, mycophenolate mofetil was reduced by 50% and prednisone to 5 mg/day, while the CNI dose was unchanged. If the viral load did not decrease within the next 4 weeks, the MMF dose was further reduced by 25% but CNI exposure remained unchanged. In the third step, if BKV viral load did not decrease or had increased within the last 4 weeks, MMF was withdrawn and replaced by everolimus (EVL) combined with a reduced CNI dose. The initial dose of everolimus was 0.25 mg twice daily the first month, then maintained at 0.75 mg twice daily in the following months; EVL trough levels were not routinely monitored due to cost issues. However, based on the study of Tedesco-Silva et al. in which KTx patients received reduced doses of cyclosporine with a reduced dose of everolimus (0.75 mg bid) at Month 6 and Month 12 post-transplantation, the patients had daily everolimus doses of 2.5 and 2.6 mg, respectively, i.e., to achieve mean everolimus trough of 5.5 ng/mL [15]. Therefore, we assume that in our study with patients having everolimus dosing of 0.75 mg bid the everolimus trough levels would have been between 3 and 4 ng/mL. In the meantime, CNI doses were concurrently reduced, aiming at trough levels of 100–150 ng/mL for cyclosporine and of 4–6 ng/mL for tacrolimus within the first month after starting everolimus.

### 2.6. Statistical Analyses

Differences between categorical variables were analyzed by the χ^2^ test, while an independent *t*-test or Mann–Whitney test was used to compare quantitative variables. All statistical analyses were conducted using the SPSS software version 20.0 (IBM, Armonk, NY, USA). Comparisons of BKV titers during follow-ups were made by repeated-measurement analysis of variance using the general linear model procedure. Logistic regression analysis combined with a log-rank test was used to calculate the factors corresponding to the decrease in BKV viremia and BKV viruria at six months compared with BKV values at the time of BKV diagnosis. A *p*-value < 0.05 was considered to represent a significant difference.

All data are available upon request to the corresponding author.

## 3. Results

### 3.1. Characteristics of the Kidney-Transplant Recipients

Of the 341 patients screened, 211 cases had at least one detection of BKV either in the urine and/or in the serum. Of the latter, only 97 (45.9%) were indicated for therapeutic intervention because of serum BKV viral load ≥ 10^4^ copies/mL and/or urinary BKV ≥ 10^7^ copies/mL (Table 3); however, only 41 patients completed the 6-month follow-up: we will mostly focus on those who were thoroughly followed-up. At the time of BKV infection diagnosis, for most cases who completed the follow-up, maintenance immunosuppression relied on tacrolimus + MMF + prednisone (92.7%). The median eGFR of the study group was 60.16 mL/min (51.83–70.62) (Table 2). Out of these 41 patients, 32 patients had BKV viremia and viruria (78.1%), nine patients had only BKV viruria (21.9%), and six cases had proven BKVN (14.6%) (Table 3). According to our immunosuppression strategy, 16 (39%) patients had only the first step, i.e., MMF dose reduction, whereas 25 (61%) required the third step, i.e., discontinuation of MMF, switch to everolimus, and CNI dose reduction (*n* = 41) (Table 4).

### 3.2. Phylogenetic Analysis of BKV Isolates

A total of 41 samples who completed the 6-month follow-up were genotyped and a phylogenetic tree was constructed from analysis of 305 bp BKV fragments, aligned with 31 reference sequences retrieved from GenBank (Figure 2). The results indicated that 29 samples (70.7%) belonged to BKV-I and 12 (29.3%) to BKV-IV, with no cases of the BKV-II or BKV-III subtypes.

The 305 bp typing-region sequences in this study and the 31 reference sequences were used to construct a neighbor-joining phylogenetic tree using Kimura’s correction.

### 3.3. Outcome of BKV Infection

At the six-month follow-up, BKV viruria, as well as BKV viremia, were significantly decreased from 9.32 log10 copies/mL to 6.09 log10 copies/mL and 3.59 log10 copies/mL to 2.45 log10 copies/mL, *p* < 0.001 and *p* = 0.024, respectively (Figure 3). At the end of follow-up, 35 out of 41 (85.4%) patients had decreased BKV viruria, and 23 out of 32 (71.8%) patients had reduced BKV viremia. Eleven out of 32 patients (34.4%) cleared BKV viremia, whereas two out of 32 (6.25%) cleared both BKV viremia and viruria. Interestingly, nine (100%) patients with only BKV viruria (mentioned in Table 4) did not develop BKV viremia after six months. In addition, the BKV titers in these nine patients tended to decrease, i.e., 10.98 log10 copies/mL vs. 7.23 log10 copies/mL, although the difference was insignificant (*p* > 0.05).

We also observed overall a significant inverse relationship between changes in eGFR and BKV viremia (r = −0.314, *p* = 0.04) (Figure 4). In contrast, there was no significant relationship between BKV viruria changes and eGFR changes (*p* = 0.97).

During the follow-up period, both tacrolimus and cyclosporine trough levels decreased; however, the changes were only statistically significant for tacrolimus (*p* = 0.008 (Figure 5). Median tacrolimus and cyclosporine trough levels at the time of BKV infection diagnosis and the end of the study (6-month follow-up) were 7.5 (3.2–15.8) and 6 (3.2–11.4) ng/mL, *p* = 0.008; and 160 (88.9–227.6) and 125 ng/mL (91.7–156.1), *p* = 0.773, respectively.

After six months, there were two cases of acute rejection and no graft loss in the BKV-infected group that did not complete the follow-up period (*n* = 56). Similarly, in the BKV-infected group with no indication for intervention (*n* = 114), there were also two cases of acute rejection and no graft loss. Within the BKV-infected patients after six months of follow-up (*n* = 41), there was no graft loss and no episodes of acute rejection. However, six patients who were converted from MMF to everolimus developed proteinuria with a concentration at the last follow-up of 0.1 g/L (0.03–0.15). When evaluating kidney graft function, overall (*n* = 41) median eGFR increased from 60.16 (27–79) to 63.94 (25–78.4) mL/min/1.73 m^2^ (*p* = 0.855), and from 62.79 (34–79) to 65.48 (38.3–78.4) mL/min/1.73 m^2^ in the BKV group (*p* = 0.695). Meanwhile, kidney allograft function worsened in the BKVN group, i.e., a decline in median eGFR from 41.32 (27.3–65) to 39.92 (25.4–61.8) mL/min/1.73 m^2^ (*p* = 0.884). However, all differences were not statistically significant (*p* > 0.05) (Figure 6).

We thereafter looked for predicting factors for BKV clearance/improvement after the first step (MMF reduction) and after the third (conversion from MMF to everolimus). We found that the patient’s age and BKV viremia load were significantly higher in the everolimus-converted group than in the MMF-reduced-dose group. In addition, the eGFR at the time of BKV infection diagnosis in the everolimus group tended to be lower than in that the MMF-reduced-dose group (Table 5).

Table 6 shows some important outcomes at 6-month follow-up, i.e., changes in BKV viremia, viruria, eGFR, and some complications according to having BKV infection with or without nephropathy. We were not able to see any differences across the two groups.

Further, we analyzed the effect of BKV genotypes, namely BKV-I and BKV-IV, on important outcomes at 6-month follow-up, i.e., changes in BKV viremia, viruria, eGFR, and some complications (acute rejection and proteinuria) (Table 7). There were no significant differences even though the rate of patients with both BKV viremia and BKV viruria reductions tended to be higher in the BKV-I group than in the BKV-IV group.

To determine the factors associated with a decrease in BKV viremia titers and BKV viruria titers, logistic regression analysis was used. The results demonstrated that no factor was found to be an independent risk factor for decreased BKV viremia and BKV viruria.

## 4. Discussion

Treating patients infected with BKV remains challenging. Because no satisfactory anti-BKV treatment is currently available, early detection of BKV infection reactivation is fundamental in preventing BKVN development. In addition, up to now almost all studies dealing with BKV infection after kidney transplantation include only patients from North America, Europe, and a few from Japan. Our study differs from the previous ones by studying BKV infection in living-kidney transplant patients from a developing country (Vietnam) with a different racial background and different BKV genotypes, i.e., BKV-I and BKV-IV. To the best of our knowledge, this is the first such study of a population in a developing country.

Some authors have previously demonstrated that early detection of BKV infection by screening urinary BKV load and serum BKV load is a practical approach for predicting kidney function deterioration [6,16]. Management approaches differ and can include dose reduction or discontinuation of the anti-metabolite, dose reduction of the calcineurin inhibitor by 25–50% targeting significantly lower levels (tacrolimus 3–4 ng/mL and cyclosporine 50–100 ng/mL, or even less), or switching from tacrolimus to cyclosporine [7,17]. The most common approach is reducing and discontinuing the anti-metabolite such as MMF. Still, a recent study suggests that both tacrolimus and cyclosporine can inhibit anti-BK T-cell responses in vitro, challenging this practice. Other treatment alternatives can include use of leflunomide, cidofovir, ciprofloxacin, rapamycin, or intravenous immunoglobulin [7,16]. However, objective data regarding BKV treatment are limited. Mammalian targets of rapamycin (mTOR) inhibitors, including sirolimus and everolimus, are widely used to avoid the nephrotoxicity of calcineurin inhibitors (CNIs) [18].

Furthermore, mTOR-inhibitor-based immunosuppression is associated with a low risk of BKV reactivation, including BKV viremia and BKVN in kidney-transplant recipients [19,20]. However, these reports are general and do not specify how the BKV viral load changes after conversion to the mTOR-inhibitor-based immunosuppression. Therefore, although mTOR-inhibitor-based therapy is likely associated with a lower risk of BKV viremia and BKVN occurrence, whether conversion to mTOR-inhibitor-based immunotherapy directly leads to a reduction in urinary BKV load and serum BKV load even in patients with a less active viral replication status requires further investigation. The objective of this study was to evaluate the effectiveness of the first regimen of MMF reduction, followed by MMF discontinuation and switch to EVL combined with CNI-dose minimization.

After six months, a significant decrease in BK viral load was found in both serum and urine (*p* = 0.02 and *p* < 0.001, respectively) (Figure 3). This result supports the assumption that reducing the dose of immunosuppressants has a role in lowering BKV load [6,7]. The results also showed that 11/32 (34.4%) patients were cleared of BKV viremia, 2/32 (6.25%) patients removed BKV in both serum and urine, and 9/9 (100%) patients who only had BKV in urine did not develop BK viremia after six months of follow-up. Almeras et al. performed simultaneous reduction of both MMF and CNI, which resulted in 8/11 patients being cleared of BKV. However, acute rejection occurred in 3/11 patients [21]. Hardinger et al. applied a procedure to discontinue MMF followed by minimization of the CNIs. The authors found that the clearance rate of BKV was 12/23, and 5/23 patients had acute rejection [22]. In this study, BKV clearance was lower than in other studies. Wojciechowski et al. conducted a pilot study to test the hypothesis that an immunosuppression-reduction strategy that includes conversion of MMF to everolimus will be more effective at treating BKV infection than MMF-dose reduction. They found that the conversion from MMF to everolimus did not improve the likelihood of achieving a 50% reduction in BKV viruria or clearance of BKV viremia after three months in kidney-transplant recipients (*p* = 0.47). Following up on the 9th and 12th months, the authors also found no significant difference between the two groups [9]. In another retrospective study, Chieh-Li Yen et al. concluded that conversion to mTOR-inhibitor-based therapy was significantly associated with a reduction in urinary BKV load [11]. Everolimus-based immunosuppressive protocol with CNI minimization and antimetabolite discontinuation effectively reduced BKV viremia in kidney transplant recipients [17]. When comparing some characteristics of the group requiring only MMF reduction with the group requiring conversion to EVL, we found that EVL could be considered in elderly patients with high BKV viremia load after stepwise removal of MMF (Table 5). However, our sample size was small. Therefore, the role of EVL in BKV virus clearance needs to be studied further.

Another critical issue in immunosuppressive regimens is the control of trough concentrations of CNIs and EVL. In this study, when patients did not respond to at least a 75% reduction in the MMF dose, the next step was to reduce the dose of CNIs and add EVL. The initial amount of EVL was 0.25 mg twice daily, then increased to 0.75 mg twice daily. There were 25/41 patients who underwent a dose reduction of CNIs in combination with EVL. During the follow-up period, both tacrolimus and cyclosporine trough levels decreased. The change was significant in the tacrolimus group, but not significant in the cyclosporine group (*p* = 0.008 and *p* = 0.773, respectively) (Figure 5). Tacrolimus and cyclosporine average trough levels at diagnosis and the end of the study were 7.5 and 6 ng/mL, and 160 and 125 ng/mL, respectively. These results were higher than in some previous studies [16,20]. However, a higher percentage of patients with decreased urinary BKV without acute rejection was recorded compared to an earlier study, i.e., 85.4% vs. 76.2%, respectively [16]. These findings suggest that the role of EVL needs to be studied more closely, as mentioned above. Nonetheless, we acknowledge that because we were not able to assess everolimus trough levels it is difficult to ascertain the overall exposure to everolimus. However, based on the study of Tedesco-Silva et al. we assume that with everolimus at 0.75 mg bid the everolimus trough levels were likely to be at 3–4 ng/mL [15].

BK virus infection can affect transplant kidney function, but does a decrease in BKV load improve kidney function? A significant inverse relationship was observed between changes in eGFR and serum BKV copies in the whole group (*n* = 41, r = −0.314, *p* = 0.046) (Figure 4). Bussalino et al. also found a strong association between the change in BKV-load-reduction status and the change in eGFR (*p* = 0.022) [20]. However, when comparing the graft kidney’s function during the follow-up, it was found that eGFR improved in the positive BKV group, but the trend was worse in the BKVN group (*p* > 0.05). We found that the eGFR of the whole group *(n* = 41) increased insignificantly after six months of follow-up (*p* > 0.05) (Figure 6). During the follow-up period of 24 months in a study focused only on patients with urinary BKV who had less severe BKV infection, Chieh-Li Yen observed a significant increase in eGFR in the EVL-converted group (*p* = 0.04) [16]. Similarly, another study showed a significant improvement in eGFR in the EVL-converted group but with a 3-year follow-up [20]. The shorter follow-up period and inadequate assessment of the histopathological damage of the transplanted kidney may be the reason for the difference between our study and the others.

BKV genotype is another risk factor for BKV-associated nephropathy. However, the role of particular genotypes or BKV variants in the development of BKVN remains unclear [23,24,25]. Therefore, we wanted to investigate further the role of genotype in the occurrence of BK-virus-associated nephropathy and the effectiveness of treatment. BKV can be classified into four genotypes (I–IV), and each genotype is further divided into several subgroups according to geographical distribution. Genotype I is the most prevalent and widespread worldwide (about 80% of reported cases), followed by genotype IV (about 15% of reported cases), mainly distributed in Europe and East Asia. In contrast, genotypes II and III are rare in all geographic regions (~5%) [26]. In this study, with 41 patients completing the analyzed follow-up period, 29 samples (70.7%) belonged to BKV-I and 12 (29.3%) to BKV-IV, with no cases of the BKV-II or BKV-III subtypes (Table 4 and Figure 2). These results are consistent with previous studies [11,25,26].

Before assessing the results, the patients were grouped according to BKV status (reactivation: yes or no) and BKV genotypes. The percentage reduction in serum and urine of BKV viral loads was compared between the positive BKV group and BKVN group and between BKV-I and BKV-IV groups. As a result, no significant difference was found between the groups (*p* > 0.05) (Table 6 and Table 7). However, this can be explained by the small sample size and short follow-up time.

One of the problems faced when reducing immunosuppressive doses to treat BKV is graft rejection. However, no case of acute rejection was observed during the study period; in addition, there was no case of allograft loss (in the 41 patients that completed the follow-up period). In the BKV-infected group with an indication for intervention that did not complete the follow-up period (*n* = 59), there were two cases of acute rejection and no graft loss. Similarly, in the BKV-infected group with no indication for intervention (*n* = 114), there were also two cases of acute rejection and no graft loss after six months. This result is different from some previous studies, which can be explained by a higher concentration of CNIs in this study compared with others, as well as the shorter follow-up time [11,16,17]. Other adverse events often associated with mTOR inhibitor therapy, including proteinuria and dyslipidemia, rarely led to everolimus discontinuation. The risk of proteinuria is higher under mTOR inhibitor therapy in patients receiving kidney transplants [10,27]. This effect is dose-dependent when everolimus is used de novo [26] and may be related to podocyte injury and inhibition of vascular endothelial growth factor signaling [27]. In this study, six cases of proteinuria were recorded in the second and third months after switching from MMF to EVL, but it decreased from 0.17 g/L (0.12–0.3) to 0.1 g/L (0.03–0.15) after reducing the dose of EVL (*p* > 0.05) and was stable with the maintenance of 0.5 mg EVL twice a day. On dyslipidemia, the data collected is insufficient to analyze and comment on.

According to previous studies, many factors affect BKV reactivation after kidney transplantation as well as BKVN development in patients after kidney transplantation [5,28]. However, the question is which factors are key to reducing BKV load during treatment. In a multivariable logistic regression analysis, we were not able to find independent risk factors. EVL was expected to be an independent risk factor for BKV load reduction when converting from MMF to EVL, but this was not observed in our study. The reason might be that in our sample, EVR was associated with reduction, discontinuation of MMF, and low CNI exposure. Thus, it cannot be ruled out that different mechanisms related to reduced CNI exposure also reduced BKV replication and improved graft function in our sample.

This study has some limitations. First, it was a short-term single-center study. Quantifying EVL concentration cannot be performed routinely in Vietnam due to the cost of testing, but it is thought that 0.5 mg EVL twice a day might be of benefit regarding BKV replication. Screening for BKV also cannot be conducted every four weeks as recommended by KDIGO guidelines because of the impact of the COVID-19 epidemic and the expensive cost compared to the average income of Vietnamese patients. Therefore, our sample size is not large enough for data analysis. Second, biopsy techniques had not been routinely performed in all patients with BKV reactivation and therefore did not accurately assess the extent of tissue damage to the transplanted kidney, and some cases with proven BKVN might have been missed. Third, no screening for de novo anti-HLA donor-specific alloantibody (DSA) was performed during and after BKV reactivation. However, clinical rejection is regarded as a clinically more relevant outcome than detection of de novo DSA. It is thought that the incidence of de novo HLA-DSA might be rather low in our cohort because no patient experienced clinical graft rejection during a follow-up of 6 months.

## 5. Conclusions

In summary, the ‘MMF-first’ immunosuppressive reduction strategy followed by discontinuing MMF, switching to EVL, and CNI minimization demonstrated efficacy in reducing viral load in kidney recipients and was associated with a low risk of clinical rejection and loss of kidney function. The therapeutic efficacy of this immunosuppression strategy was not different in patients with genotype I and genotype IV. Thus, prospective randomized trials with larger sample sizes and longer follow-up times are needed to compare the risks and benefits of immunosuppressive strategies in transplant recipients with high BKV loads.

## Figures and Tables

**Figure 1 jcm-11-07297-f001:**
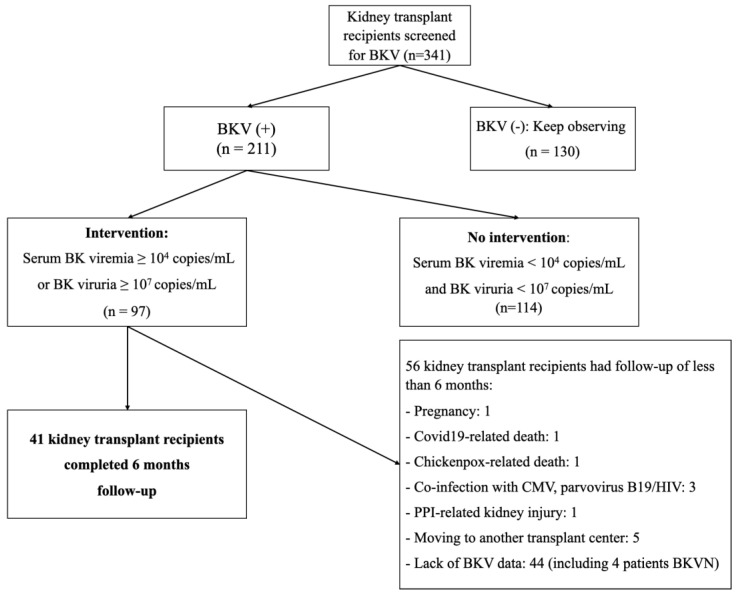
Study flow chart. Abbreviations: CMV, cytomegalovirus; HIV, human immunodeficiency virus; PPI, proton pump inhibitor.

**Figure 2 jcm-11-07297-f002:**
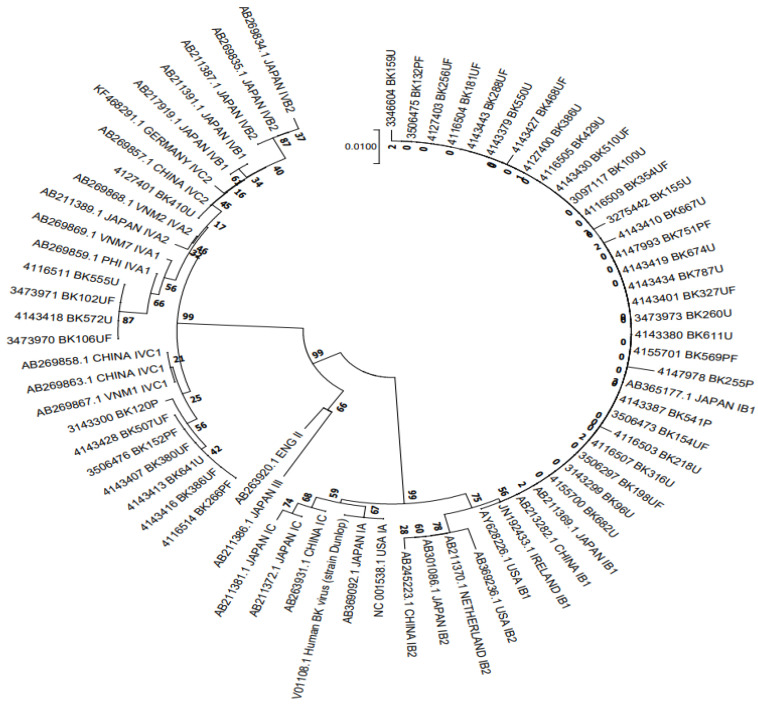
NJ phylogenetic tree clustering of 29 BKV-I and 12 BKV-IV polyomavirus sequences to determine subgroups (*n* = 41).

**Figure 3 jcm-11-07297-f003:**
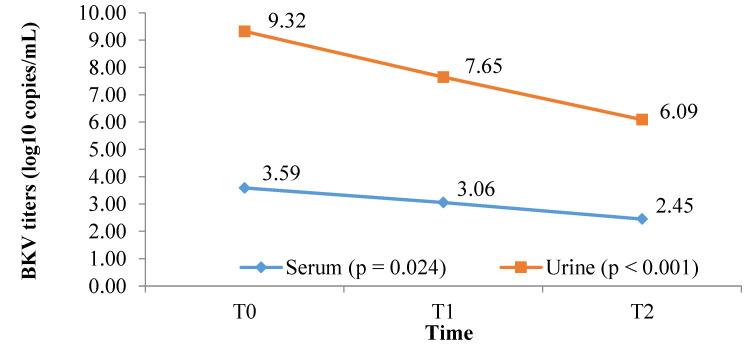
BKV viremia and BKV viruria titers during the 6-month follow-up period in 41 patients who completed follow-up time (*p* = 0.024 for serum; *p* < 0.001 for urine; both ANOVA for repeated measures).

**Figure 4 jcm-11-07297-f004:**
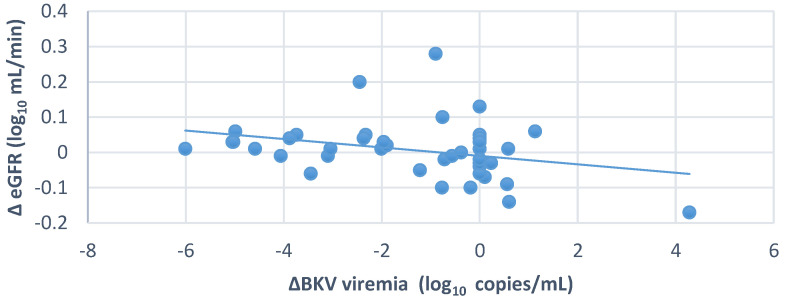
Relationship between changes in eGFR and changes in BKV viremia (*n* = 41 KTx patients); r = −0.314; *p* = 0.04.

**Figure 5 jcm-11-07297-f005:**
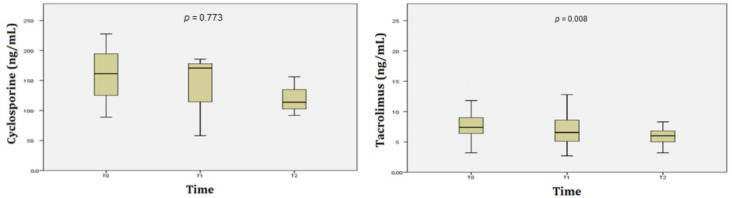
Cyclosporine and tacrolimus trough levels during the 6-month follow-up period in 41 KTx patients with BKV infection.

**Figure 6 jcm-11-07297-f006:**
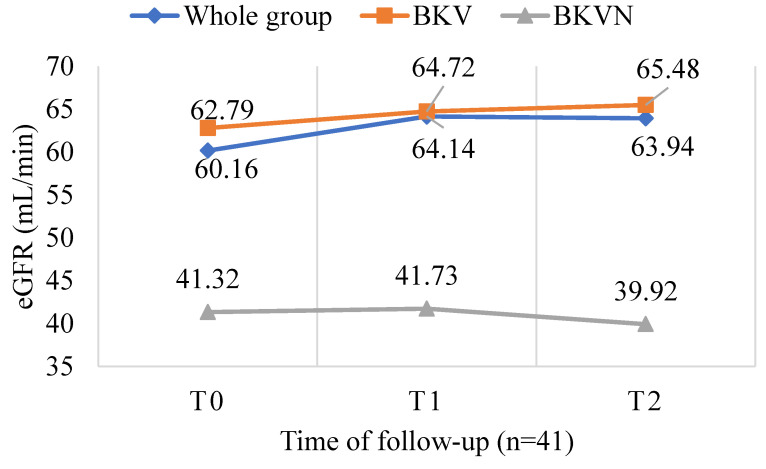
Changes in eGFR during the 6-month follow-up period in 41 KTx patients with BKV infection with *p* = 0.855 for the whole group (*n* = 41), *p* = 0.695 for the BKV group (*n* = 35), and *p* = 0.884 for the BKVN (*n* = 6).

**Table 1 jcm-11-07297-t001:** Primer sequences used in our study.

Primers	Sequences	Position	Size	References
BKV_S	ATC AAA GAA CTG CTC CTC AAT	1361–1381	580 bp	[13]
BKV_AS	GCA CTC CCT GCA TTT CCA AGGG	1919–1940		
BKV-F	CAA GTG CCA AAA CTA CTA AT	1630–1649	327 bp	[14]
BKV-R1	TGC ATG AAG GTT AAG CAT GC	1937–1956		

Abbreviations: BKV, BK polyomavirus; bp, base pairs. With respect to GeneBank accession number V01108 (Dunlop strain).

**Table 2 jcm-11-07297-t002:** Demographic, clinical, and transplant-related data at the time of BKV or BKVN diagnosis.

Parameters		Whole Group (*n* = 341)	BKV Infection Group without Indication for Intervention (*n* = 114)	BKV Infection Group with an Indication for Intervention (*n* = 97)	BKV Infection Group with 6-Month Follow-Up (*n* = 41)
Age, years (mean ± SD)		38.2 ± 11.2	37.2 ± 9.6	37.9± 10.7	37.4 ± 10.5
Male, *n* (%)		242 (70.9)	82 (71.9)	64 (65.9)	29 (79.7)
Times of HD, months, median (IQR)		9.3 (2–116)	4 (0–120)	12.6 (3–45)	12 (4–36)
Times after KTx to the initial diagnosis of BKV, months, median (IQR)		10.02 (1–120)	12.36 (1–120)	10.32 (1–100)	7.32 (1–36)
Pre-transplant ESKD treatment:					
Peritoneal dialysis, *n* (%)		10 (2.9)	2 (1.8)	3 (3.1)	1 (2.4)
Hemodialysis, *n* (%)		299 (87.7)	99 (86.8)	88 (90.7)	37 (90.3)
No, *n* (%)		32 (9.4)	13 (11.4)	6 (6.2)	3 (7.3)
Diabetes before KTx, *n* (%)		9 (2.6)	2 (1.75)	2 (2.1)	1 (2.4)
Living donor, *n* (%)		341 (100)	114 (100)	97 (100)	41 (100)
Living Donor *n* (%)	Related	41 (12.02)	4 (3.51)	10 (10.3)	4 (9.75)
	Unrelated	300 (87.98)	110 (96.49)	87 (89.7)	37 (80.25)
Donor age, years, (mean ± SD)		33.5 ± 9.5	34.05 ± 7.8	33.18 ± 8.25	31.1 ± 6.02
Donor age > 60, years, *n* (%)		8 (2.3)	2 (1.75)	1 (1.03)	0 (0)
Female donors, *n* (%)		94 (27.6)	30 (26.3)	25 (25.8)	13 (31.7)
HLA mismatches ≥ 1, *n* (%)		337 (98.8)	105 (92.1)	96 (98.9)	41 (100)
Allele-HLA mismatches ≥ 1, *n* (%)	A	270 (79.2)	80 (70.1)	87 (89.6)	34 (82.9)
	B	295 (86.5)	82 (71.9)	88 (90.7)	39 (95.1)
	DRB1	265 (77.7)	88 (77.2)	84 (86.6)	34 (82.9)
Positive PRA before KTx, *n* (%)		69 (20.2)	27 (23.7)	16 (16.5)	8 (19.5)
Induction therapy before KTx	Basiliximab induction, *n* (%)	335 (98.2)	109 (95.6)	96 (98.9)	40 (97.6)
	ATG induction, *n* (%)	6 (1.8)	5 (4.4)	1 (1.1)	1 (2.4)
IS regimen	Cyclosporine + MMF+ Prednisolone, *n* (%)	39 (11.4)	21 (18.5)	13 (13.4)	3 (7.3)
	Tacrolimus +MMF + Prednisolone, *n* (%)	302 (89.6)	93 (81.5)	84 (86.6)	38 (92.7)
Serum creatinine at the time of BKV diagnosis, median (IQR), µmol/L		107.3 (88.2–124.8)	110.8 (66–126.2)	111.3 (90.2–129.4)	112.2 (92.8–128.8)
Proteinuria at the time of BKV diagnosis (Yes), *n* (%)		31 (9.1)	5 (4.4)	8 (8.2)	0 (0)
eGFR (mL/min) at the time of BKV diagnosis, median (IQR)		64.5 (53.8–76.3)	62.4 (55.5–71.9)	61.32 (51.4–72.1)	60.16 (51.8–70.6)

Abbreviations: ESKD, end-stage kidney disease; KTx, kidney transplantation; IS, immunosuppression; SD, standard deviation; eGFR, estimated glomerular filtration rate; CNI, calcineurin inhibitor; BKV, BK virus; MMF, mycophenolate mofetil; HD, hemodialysis; ATG, anti-thymocyte globulin; PRA, panel reactive alloantibodies.

**Table 3 jcm-11-07297-t003:** Characteristics of BKV virus load in serum and urine of the whole group (*n* = 341).

Parameters		BKV Viruria, Copies/mL *n* (%)			Total *n* (%)
		<250 (*)	250−10^7^	≥10^7^	
BKV viremia, copies/mL *n* (%)	<250 *	130 (72.6)	38 (21.2)	11 (6.2)	179 (52.5)
	250–10^4^	21 (20.8)	55 (54.5)	25 (24.7)	101 (29.6)
	≥10^4^	0 (0)	15 (24.5)	46 (74.5)	61 (17.9)
Total, *n* (%)		151 (44.3)	108 (31.7)	82 (24)	341 (100)
BKV viremia ≥ 10^4^ and/or BKV viruria ≥ 10^7^ copies/mL, *n* (%)					97 (28.4)

*: under the cut-off BK viral load detection (250 copies/mL) which means negative BKV diagnosis.

**Table 4 jcm-11-07297-t004:** Characteristics of BKV virus load in serum and urine according to immunosuppressive strategies.

Parameters		BKV Infection Group without Indication for Intervention (*n* = 114)	BKV Infection Group with an Indication for Intervention (*n* = 97)	BKV Infection Group with 6-Month Follow-Up (*n* = 41)
BKV genotypes, *n* (%)	Genotypes			
	I	31 (57.4)	59 (61.5)	29 (70.7)
	IV	23 (42.6)	37 (38.5)	12 (29.3)
BKV viremia *n* (%)	<250 * copies/mL	37 (32.5)	12 (12.3)	9 (21.9)
	250–10^4^ copies/mL	77 (67.5)	24 (24.7)	15 (36.6)
	≥10^4^ copies/mL	0 (0)	61 (63)	17 (41.5)
BKV viruria *n* (%)	<250 * copies/mL	24 (21.1)	0 (0)	0 (0)
	250–10^7^ copies/mL	90 (78.9)	17 (17.5)	5 (12.2)
	≥10^7^ copies/mL	0 (0)	80 (82.5)	36 (87.8)
BKV viremia ≥ 10^4^ and/or BKV viruria ≥ 10^7^ copies/mL, *n* (%)		0 (0)	97 (100)	41 (100)
BKVN, *n* (%)		0 (0)	10 (10)	6 (14.6)
Treatment strategies for BKV/BKVN, *n* (%)	MMF reduction only	-	-	16 (39)
	MMF reduction, then discontinuation and conversion to EVL with CNI minimization	-	-	25 (61)

(*): under the cut-off BK viral load detection (250 copies/mL) which means negative BKV diagnosis. Abbreviations: CNI, calcineurin inhibitor; BKV, BK virus; BKVN, BKV nephropathy; MMF, mycophenolate mofetil; EVL, everolimus.

**Table 5 jcm-11-07297-t005:** Outcome of BKV infection after 6-month follow-up according to immunosuppressive strategies.

Parameters	MMF-Reduced Dose (*n* = 16)	Conversion from MMF to EVL (*n* = 25)	*p*-Value
Recipients age, years, median (IQR)	31.5 (29.0, 38.5)	41.0 (34.0, 47.0)	*0.01*
Recipient Gender Female Male	4 (25.0%)	8 (32.0%)	0.63
	12 (75.0%)	17 (68.0%)	
HLA-A, B, DR matching, n, median (IQR)	3.0 (2.0, 3.5)	3.0 (2.0, 3.0)	0.30
HLA-A, B, DR mismatches, n, median (IQR)	3.0 (2.5, 4.0)	3.0 (3.0, 4.0)	0.30
PRA before KTx Negative Positive	15 (93.8%)	18 (72.0%)	0.08
	1 (6.3%)	7 (28.0%)	
Dose of TacT0, mg/d, median (IQR)	6.0 (5.0, 7.5)	6.0 (5.0, 6.5)	0.39
Dose of TacT1, mg/d, median (IQR)	5.3 (4.3, 6.5)	4.8 (3.5, 6.0)	0.21
Dose of TacT2, mg/d, median (IQR)	5.0 (4.0, 6.5)	4.5 (3.0, 6.0)	0.31
Dose of MMF T0, mg/d, median (IQR)	2000 (1500, 2000)	2000 (1500, 2000)	-
Dose of MMF T1, mg/d, median (IQR)	1000 (750, 1000)	0 (0, 1000)	-
Dose of MMF T2, mg/d, median (IQR)	750 (500, 750)	0 (0, 0)	-
Dose of EVL, mg/d, median (IQR)	-	0.5 (0.5, 1.25)	-
Dose of steroid T0, mg/d, median (IQR)	5 (5,10)	5(5, 10)	-
Dose of steroid T1, mg/d, median (IQR)	5	5	-
Dose of steroid T2, mg/d, median (IQR)	5	5	-
Tac level T0, ng/mL, median (IQR)	8.4 (7.4, 10.1)	6.5 (5.6, 8.2)	0.01
Tac level T1, ng/mL, median (IQR)	7.1 (5.3, 9.1)	5.8 (4.8, 8.6)	0.38
Tac level T2, ng/mL, median (IQR)	6.4 (5.5, 7.2)	5.4 (5.0, 6.5)	0.06
BKV viremia T0, median (IQR)	3.07 (0–3.98)	4.12 (2.99–5.09)	0.02
BKV viremia, T2, median (IQR)	0 (0–3.29)	3.19 (0–4.14)	0.02
BKV viruria T0, median (IQR)	9.54 (8.41–10.38)	9.15 (7.86–10.23)	0.65
BKV viruria T2, median (IQR)	5.32 (3.57–7.93)	7.41 (2.88–8.18)	0.50
eGFR T0, median (IQR) (mL/min)	63.6 (55.0, 70.2)	58.5 (47.3, 70.0)	0.45
eGFR T1, median (IQR) (mL/min)	65.3 (56.8, 70.0)	62.2 (53.0, 67.9)	0.17
eGFR T2, median (IQR) (mL/min)	67.8 (61.1, 74.1)	57.9 (44.6, 66.6)	0.01

Abbreviations: MMF, mycophenolate mofetil; EVL, everolimus; KTx, kidney transplantation; PRA, panel reactive alloantibodies; Tac, tacrolimus; BKV, BK virus; eGFR, estimated glomerular filtration rate; T, time, e.g., T0 corresponds to the time at which the clinical intervention regarding BKV infection was implemented.

**Table 6 jcm-11-07297-t006:** Six-month follow-up outcomes in BKV-infected patients with or without BKVN.

Some Characteristics		BKV (+) (*n* = 35)	BKVN (*n* = 6)	*p*-Value
Changes in BKV viremia	No change, *n* (%)	9 (25.7)	0 (0)	0.56
	Decrease, *n* (%)	19 (54.3)	4 (66.7)	
	Increase, *n* (%)	7 (20)	2 (33.3)	
Changes in BKV viruria	No change, *n* (%)	0 (0)	0 (0)	0.81
	Decrease, *n* (%)	30 (85.7)	5 (83.4)	
	Increase, *n* (%)	5 (14.3)	1 (16.7)	
Change of eGFR	Increase, *n* (%)	21 (60)	3 (50)	0.67
	Decrease, *n* (%)	14 (40)	3 (50)	
Complications	Acute rejection, *n* (%)	0 (0)	0(0)	
	Proteinuria (+), *n* (%)	3 (8.57)	3 (50)	0.57

Abbreviations: BKV, BK virus; BKVN, BKV nephropathy; eGFR, estimated glomerular filtration rate.

**Table 7 jcm-11-07297-t007:** Six-month follow-up outcomes in BKV-infected patients according to BKV genotype.

Some Characteristics		BKV-I (*n* = 29)	BKV-IV (*n* = 12)	*P*
The changes in serum BKV load	No change, *n* (%)	7 (24.1)	2 (16.7)	0.63
	Decrease, *n* (%)	15(51.7)	8(66.6)	
	Increase, *n* (%)	7 (24.1)	2 (16.7)	
The changes in urine BKV load	No change, *n* (%)	0 (0)	0(0)	0.11
	Decrease, *n* (%)	23(79.3)	12(100)	
	Iincrease, *n* (%)	6(20.7)	0(0)	
Change in eGFR	Increase, *n* (%)	17 (58.6)	7 (58.3)	1.00
	Decrease, *n* (%)	12 (41.4)	5 (41.7)	
Complications	Acute rejection, *n* (%)	0(0)	0(0)	
	Proteinuria (+), *n* (%)	4 (13.8)	2 (16.7)	0.62

Abbreviations: BKV, BK virus; BKVN, BKV nephropathy; eGFR, estimated glomerular filtration rate.

## Data Availability

Data are available upon reasonable request.

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
