# Peer review of "Stepwise Reduction of Mycophenolate Mofetil with Conversion to Everolimus for the Treatment of Active BKV in Kidney Transplant Recipients: A Single-Center Experience in Vietnam"

_jcm, 2022, doi:10.3390/jcm11247297_

Round 1

Reviewer 1 Report

I have read an article titled “Stepwise reduction of MMF with conversion to EVL for the treatment of active BKV in kidney recipients: A single-center experience in Vietnam”. My main concerns are about the novelty and the interpretation of the study. Below are my points regarding the manuscript.

1.      Usually, the title should not contain abbreviation but I will leave the decision to adjust the title to the authors and the journal. Also, the authors may consider to use “kidney transplant recipients” instead of “kidney recipients”.

2.      In most of the previously reported studies, the prevalence of detectable BKV was around 30%. I found that the prevalence of detectable BKV replication in this study was surprisingly high (62%). When I looked at the ref.2 where the authors cited for the prevalence of BKV, it seems to be an incorrect reference. Could the authors provide the correct reference that actually reported the prevalence of BK viuria of 60% as in line 58?

3.      What was the protocol that the center used to screening for active BKV replication? This is important because if there was no standard protocol (like every 2 months until 12 months), there would be a high chance to missed some cases and non-cases and this might lead to miss-interpretation of the prevalence in the study, also, the true outcomes of BKV infection.

4.      Please provide the time and duration for the inclusion of these 344 kidney transplant recipients.

5.      Line 140, I believe the Banff classification was not reported in 2018 (only a reference guide that was published in Transplantation journal). Could the author provide the proper citation for the Banff classification used in this study?

6.      Line 166: what is the meaning of *?

7.      Table 3: It seems interesting that 3 significant (>104) BK viremic patients did not have BK viuria, can the authors describe the characteristic of these patients? Maybe in a supplementary material. For example, does the urine and blood were tested in the same time?

8.      For the information in the method section, some recipients were treated based on BK viuria >107 only, without the testing of BK viremia. This might cause a bias in the interpretation of the outcomes of infection since a substantial proportion of patients with BK viuria do not have BK viremia/BKVAN or other negative consequences. And not uncommon that BK viuria (without BK viremia) will spontaneously resolved without the need for interventions. The discussion in line 315-317 might not be appropriate without mentioning this limitation.

9.      Were all 9 patients (line 176) had only BK viuria because the blood was not tested, or was it tested negative?

10.   Several typo errors of the symbols, superscripts, etc. need corrections.

11.   I do not agree with line 336-340. Since the study was conducted as the step-wise fashion, it cannot be interpreted that EVL conversion should be considered first in the elderly and high BK viremic patients. It is still possible that, without MMF-reduction before, the EVL-conversion only will not work. The study was not designed to compare the efficacy of two interventions and the characteristics of the patients.

12.   The lacking of EVL concentration was one of the main limitation and that make the study less interpretable regarding the true efficacy/toxicity of this strategy.

13.   Could the authors present the novelty of this study in one short paragraph/sentences in the discussion?

Author Response

Comment 1. Usually, the title should not contain abbreviation but I will leave the decision to adjust the title to the authors and the journal. Also, the authors may consider to use “kidney transplant recipients” instead of “kidney recipients”.

Response: We agree with your comment and edited it accordingly. The title becomes:

 “Stepwise reduction of mycomepholate mofetil with conversion to everolimus for the treatment of active BKV in kidney transplant recipients: A single-center experience in Vietnam”

Comment 2. In most of the previously reported studies, the prevalence of detectable BKV was around 30%. I found that the prevalence of detectable BKV replication in this study was surprisingly high (62%). When I looked at the ref.2 where the authors cited for the prevalence of BKV, it seems to be an incorrect reference. Could the authors provide the correct reference that actually reported the prevalence of BK viuria of 60% as in line 58?

Response: The rate of BKV infection depends on many factors, including the study design and the PCR technique's cut-off point at each center. We thank you for spotting our reference citation error: we corrected it. In addition, we added 2 other references.

  1. Comerlato, J., et al., Molecular detection and characterization of BK and JC polyomaviruses in urine samples of renal transplant patients in Southern Brazil. J Med Virol, 2015. 87(3): p. 522-8.
  2. Yoon, S.H., et al., Clinical impact of BK virus surveillance on outcomes in kidney transplant recipients. Transplant Proc, 2015. 47(3): p. 660-5.
  3. Sharma, R. and M. Zachariah, BK Virus Nephropathy: Prevalence, Impact and Management Strategies. Int J Nephrol Renovasc Dis, 2020. 13: p. 187-192.

Comment 3. What was the protocol that the center used to screening for active BKV replication? This is important because if there was no standard protocol (like every 2 months until 12 months), there would be a high chance to miss some cases and non-cases and this might lead to miss-interpretation of the prevalence in the study, also, the true outcomes of BKV infection.

Response: The guidelines recommend frequent checking for BKV viral DNA in urine and/or blood for the initial 1-2 years after kidney transplantation. However, polymerase chain reaction (PCR)-based screening assays are expensive. Quantitative BKV PCR costs around 50- 60 US dollars (USD) per assay in Vietnam. Surveillance of a patient in the first post-transplant year, as per KDIGO guidelines, would cost around 400-600 USD. Patients undergoing kidney transplantation in public hospitals in middle- and low-income countries like Vietnam have a poor socio-economic background. Because they have limited funding and this test is not covered by medical insurance, we are unable to screen them routinely for BKV with expensive investigations like PCR-based assays as per guidelines. We therefore mentioned this in the limitations section of the study on page 16, lines 436-438, as follows.

    “Screening for BKV also cannot be done every four weeks as recommended by KDIGO guidelines because of the impact of the covid-19 epidemic and the expensive cost compared to the average income of Vietnamese patients”

Comment 4. Please provide the time and duration for the inclusion of these 344 kidney transplant recipients.

Response: Thank you for your comment. We have added these data according to your suggestion in Table 2.  Overall, the mean time from the kidney transplant to the first diagnosis of BKV infection was 10.02 (1-120) months.

Comment 5: Line 140, I believe the Banff classification was not reported in 2018 (only a reference guide that was published in Transplantation journal). Could the author provide the proper citation for the Banff classification used in this study?

Response: Thank you for your valuable comment. We have checked again and corrected this reference: “BK virus-associated nephropathy was confirmed using immunohistochemically nuclear staining with anti-SV40 antibody. Concomitant acute rejection was evaluated by using Banff 2017 scoring system”.

Comment 6: Line 166: what is the meaning of *?

Response: Thank you for your comment. We double-checked and determined this was a typo error. We have removed this notation in this manuscript, line 169.

Comment 7: Table 3: It seems interesting that 3 significant (>104) BK viremic patients did not have BK viuria, can the authors describe the characteristic of these patients? Maybe in a supplementary material. For example, does the urine and blood were tested in the same time?

Response: Thank you for your valuable comment. We doubled checked and found that these 3 patients had NO BKV in the urines. Therefore, these three patients were removed from the study. The total number of patients for this study was 341 instead of 344, and accordingly we reanalyzed the relevant data sheets. This only affects section 3.1 on the general characteristics of studied subjects. Since the three patients excluded from the study were all in the group that did not complete the 6-month follow-up period, the analysis results in the 41 patients who completed the study were unaffected. We are very sorry about this issue.

Comment 8: For the information in the method section, some recipients were treated based on BK viuria >107 only, without the testing of BK viremia. This might cause a bias in the interpretation of the outcomes of infection since a substantial proportion of patients with BK viruria do not have BK viremia/BKVAN or other negative consequences. And not uncommon that BK viruria (without BK viremia) will spontaneously resolved without the need for interventions. The discussion in line 315-317 might not be appropriate without mentioning this limitation.

Response: Thank you for your comment. In this study, we performed BKV virus testing in all patients in both blood and urine. We have added this information to the methods section on page 2, lines 84-86. Before the study submitted to JCM, we had another study to evaluate the role of urinary BKV viral load in the prognosis of BKV-associated nephropathy: this previous work is under consideration for publication in another journal (see appendix to this Point by point doclment). Briefly, we found that the urinary viral load of BKV with a cut-off point of 8.89 x 108 copies/ml had a predictive value of BKVN, and the sensitivity and specificity were 87.5% and 93.6%, respectively. Based on these results, and because maintaining functional a LIVING-kidney transplant is of outmost importance in Vietnam, we intervene earlier, i.e., when the urinary BKV load is greater than 107 copies/ml in order to limit the rate of BKVN. We have provided some more background information on that study in the appendix.

Comment 9: Were all 9 patients (line 176) had only BKV viuria because the blood was not tested, or was it tested negative?

Response: Thank you for your comment. In this study, we performed BKV virus testing in all patients in both blood and urine. These nine patients only had BKV in the urine, i.e., BKV viremia was negative.  We have added this information to the Methods’section on page 2, lines 84-86.

Comment 10: Several typo errors of the symbols, superscripts, etc. need corrections.

Response: Thank you for your comment. We have accordingly made the corrections.

Comment 11: I do not agree with line 336-340. Since the study was conducted as the step-wise fashion, it cannot be interpreted that EVL conversion should be considered first in the elderly and high BK viremic patients. It is still possible that, without MMF-reduction before, the EVL-conversion only will not work. The study was not designed to compare the efficacy of two interventions and the characteristics of the patients.

Response: Thank you for your comment. We have revised it according to your suggestion, lines 345-349 as follows:

“When comparing some characteristics of the group requiring only MMF reduction with the group requiring conversion to EVL, we found that EVL could be considered in elderly patients with high BKV viremia load after stepwise removal of MMF (Table 5). However, our sample size was small. Therefore, the role of EVL in BKV virus clearance needs to be studied further”.

Comment 12: The lacking of EVL concentration was one of the main limitation and that make the study less interpretable regarding the true efficacy/toxicity of this strategy.

Response: Thank you for your comment. We also found that the lacking of EVL concentration was one of the main limitations of the study. This is now mentioned in the discussion as study limitation.

Comment 13: Could the authors present the novelty of this study in one short paragraph/sentences in the discussion?

Response: Thank you for your comment: we added the following sentences “Up to now almost all studies dealing with BKV infection after kidney transplantation are including patients from Nort-America, Europe and a few from Japan. Our study differs from the previous ones by studying BKV infection in living-kidney transplant patients from a developing country (Vietnam) with a different racial background and different BKV genotypes, i.e., BKV-I and BKV-IV genotypes (see lines 292-6). Also, in the discussion section, lines 394-399 we added:

 “Before assessing the results, the patients were grouped according to BKV status (reactivation: yes or no) and to BKV genotypes. The percentage reduction in serum and urine of BKV viral loads was compared between the positive BKV group and BKVN group and between BKV-I and BKV- IV groups. As a result, no significant difference was found between the groups (p>0.05) (Tables 6, and 7). However, this can be explained by the small sample size and short follow-up time.”

Reviewer 2 Report

This manuscript addresses a clinically important topic and presents the findings in a well-written manner. I thank the authors for the opportunity to review it.

Generally, the paper should be reviewed by a native English speaker to enhance its readability. The authors sometimes use "BKV virus"; since the V stands for virus, this is not needed.

The introduction should be expanded. Some of the information included in the discussion section belongs in the introduction such as the risks of reducing MMF dosage and mTOR usage and the risks of rejection with reduction of immunosuppressive medication doses. 

The Methods section is well written and provides appropriate detail. 

The Discussion section is well written; as detailed above, some sections would be better served in the Introduction.

Author Response

Comment 1: Generally, the paper should be reviewed by a native English speaker to enhance its readability. The authors sometimes use "BKV virus"; since the V stands for virus, this is not needed.

Response: We agree with your comments and we have made it revised accordingly.

Comment 2: The introduction should be expanded. Some of the information included in the discussion section belongs in the introduction such as the risks of reducing MMF dosage and mTOR usage and the risks of rejection with reduction of immunosuppressive medication doses. 

Response: We added some of the information as you suggested. These informations are now presented in the section introduction, lines 74-77.

“However, a risk of rejection may arise when reducing the dose of immunosuppressants [8], [9]. EVL is thought to play a key role in BKV control; however, its use may increase the risk of proteinuria and dyslipidemia [7], [10]”.

Comment 3:  The Discussion section is well written; as detailed above, some sections would be better served in the Introduction.

Response: Thank you for your valuable comment. We have taken some informations from the discussion to add them into the introduction to clarify further the risks associated with reducing the dose of immunosuppressants (see lines 74-77).

Reviewer 3 Report

The authors present a local perspective at the BKV nephropathy in kindey transplant recipients. The study could be improved by adding some more data. The authors should provide information on MMF inital dosage and its changes after the intervention, steroid dose should also be presented. Data on time after transplant to the initial diagnosis of BKV viremia/viruria should be presented. As these are all living donors transplantaion data on donor-recipient relations should be added and HLA mismatching described in more detail - which loci were analysed, how many recipients had haploidentical donor. The lack of everolimus trough levels is a drawback - I would suggest looking for a cooperation partner to do the measurements, if the authors have the samples stored.

The observation period seems very short to have strong conclusions. I would suggest keeping these patients in observation and reporting the data with longer observation time. With this strategy the most common complication to be expected is chronic rejection. This is much more difficult to diagnose based only on clinical data without the biopsy. De novo DSA may be treated as a surrogate marker for chronic rejection but this data is also lacking. Again, if the authors have the samples they should look for a cooperation partner to improve their study.

Author Response

Comment 1: The authors should provide information on MMF inital dosage and its changes after the intervention, steroid dose should also be presented. Data on time after transplant to the initial diagnosis of BKV viremia/viruria should be presented.

Response: Thank you for your valuable comment. We have added these data according to your suggestion in Table 5. 

Comment 2: As these are all living donors transplantaion data on donor-recipient relations should be added and HLA mismatching described in more detail - which loci were analysed, how many recipients had a haploidentical donor.

Response: Thank you for your valuable comment. We have added these data according to your suggestion in Table 2.  In Vietnam, regarding HLA typing we only assess loci A, B and DRB1. Regarding the relationship between the donors and the recipients, 12.02% are living-related, whereas the others are living-unrelated. All the donors and recipients are registered through the national organ coordination center and strictly regulated by law.

Comment 3: The lack of everolimus trough levels is a drawback - I would suggest looking for a cooperation partner to do the measurements, if the authors have the samples stored.

Response: Thank you for your valuable comment. In Vietnam, very few kidney transplant patients are on EVL-based immunosuppression. In addition, everolimus trough level measurements are not reimbursed by the health care system. As a consequence, patients refuse to assess EVL trough levels. Finally, we do not have for these patients stored frozen blood samples. Indeed, we have mentioned this drawback as a limitation of our study.

Comment 4: The observation period seems very short to have strong conclusions. I would suggest keeping these patients in observation and reporting the data with longer observation time. With this strategy the most common complication to be expected is chronic rejection. This is much more difficult to diagnose based only on clinical data without the biopsy. De novo DSA may be treated as a surrogate marker for chronic rejection but this data is also lacking. Again, if the authors have the samples they should look for a cooperation partner to improve their study.

Response: Thank you for your valuable comment. We have also mentioned this short time study period as a limitation of our study. The treatment and follow-up of patients after transplantation are our daily work. Therefore, all these patients will be followed up by us, and longer-term studies will be published. We hope that in the future, testing techniques, as well as medical insurance, will support more biological tests that are otherwise done in western countries for that type of patients. We appreciate your kindness and hope to receive your help in the future in other research.

Round 2

Reviewer 1 Report

The lack of everolimus concentration monitoring was an important obstacle for the interpretation of the treatment efficacy. The good side in this study was the phylogenetic information of BKV in South-East Asia. However, low number of the complete-follow-up patients (that were included to the final analyses) was also an important bias which the true prevalence of BKV phylogenetics could not be confidently elucidated. 

Author Response

The lack of everolimus concentration monitoring was an important obstacle for the interpretation of the treatment efficacy. We totally agree. However, based on the results of the study of Tedesco-Silva Jr et al. (now reference 15) we assume that with an everolimus daily dose of 1.5 mg the everolimus trough levels would have been at 3-4 ng/mL, i.e., a concentration that retains immunosuppressive effects as well as anti-viral effects when associated with reduced doses of CNIs.

The good side in this study was the phylogenetic information of BKV in South-East Asia. We agree.

However, low number of the complete-follow-up patients (that were included to the final analyses) was also an important bias which the true prevalence of BKV phylogenetics could not be confidently elucidated. We agree but we can not change the figures.

Reviewer 2 Report

The manuscript is greatly improved from the original submission. I would still prefer to see some of the discussion moved to the introduction but this is a stylistic issue not a higher priority and should not prevent publication.

Author Response

We warmly thank Reviewer 2 and we are pleased that we addressed all his/her comments/concerns.

Reviewer 3 Report

My comments are addressed.

Author Response

We warmly thank Reviewer 3 that endorses our manuscript.